# Dysregulation of Neuronal Genes by Fetal-Neonatal Iron Deficiency Anemia Is Associated with Altered DNA Methylation in the Rat Hippocampus

**DOI:** 10.3390/nu11051191

**Published:** 2019-05-27

**Authors:** Yu-Chin Lien, David E Condon, Michael K Georgieff, Rebecca A Simmons, Phu V Tran

**Affiliations:** 1Center for Research on Reproduction and Women’s Health, Perelman School of Medicine, The University of Pennsylvania, Philadelphia, PA 19104, USA; ylien@pennmedicine.upenn.edu (Y.-C.L.); dec986@gmail.com (D.E.C.); 2Department of Pediatrics, University of Minnesota School of Medicine, Minneapolis, MN 55455, USA; georg001@umn.edu; 3Children’s Hospital of Philadelphia, Philadelphia, PA 19104, USA

**Keywords:** hippocampus, DNA methylation, DNA sequencing, iron, neurobiology, transcriptome, micronutrient deficiency, neuroplasticity

## Abstract

Early-life iron deficiency results in long-term abnormalities in cognitive function and affective behavior in adulthood. In preclinical models, these effects have been associated with long-term dysregulation of key neuronal genes. While limited evidence suggests histone methylation as an epigenetic mechanism underlying gene dysregulation, the role of DNA methylation remains unknown. To determine whether DNA methylation is a potential mechanism by which early-life iron deficiency induces gene dysregulation, we performed whole genome bisulfite sequencing to identify loci with altered DNA methylation in the postnatal day (P) 15 iron-deficient (ID) rat hippocampus, a time point at which the highest level of hippocampal iron deficiency is concurrent with peak iron demand for axonal and dendritic growth. We identified 229 differentially methylated loci and they were mapped within 108 genes. Among them, 63 and 45 genes showed significantly increased and decreased DNA methylation in the P15 ID hippocampus, respectively. To establish a correlation between differentially methylated loci and gene dysregulation, the methylome data were compared to our published P15 hippocampal transcriptome. Both datasets showed alteration of similar functional networks regulating nervous system development and cell-to-cell signaling that are critical for learning and behavior. Collectively, the present findings support a role for DNA methylation in neural gene dysregulation following early-life iron deficiency.

## 1. Introduction

Fetal and neonatal (early-life) iron deficiency with or without anemia affects more than 30% of pregnant women and preschool age children worldwide, and results in long-term cognitive and behavioral abnormalities [1,2,3,4,5,6,7,8]. We have previously investigated the effects of early-life iron deficiency using a rat model, whereby pups were made iron-deficient (ID) from gestational day 2 through postnatal day (P) 7 by providing pregnant and nursing dams with an ID diet, after which they were rescued with an iron-sufficient (IS) diet. This model of maternal-fetal iron deficiency results in a 50% reduction in brain iron concentration by P7 [9], the age at which rat brain development approximates that of a full-term human newborn [10,11]. The deficit in brain iron content is similar to the degree of brain iron deficiency observed in full-term newborn humans [12,13]. Similar cognitive and behavioral abnormalities are observed in our rat model [14,15,16] and are accompanied by abnormal neuronal morphology [17,18] and glutamatergic neurotransmission [19] in the hippocampus. Iron treatment starting at P7 resolves brain iron deficiency by P56 [20]. Despite this resolution, the formerly iron-deficient (FID) rats show persistent cognitive impairment accompanied by abnormal neuronal morphology [17,18], glutamatergic neurotransmission [19], and lower expression of genes critical for neural plasticity in the hippocampus [21,22,23]. The persistent dysregulation of hippocampal gene expression in the adult FID rat hippocampus [22] suggests a possible role for epigenetic regulation. Indeed, in a previous study we showed that early-life iron deficiency induced epigenetic modifications at the *Bdnf* locus, a critically important gene coding for a growth factor that regulates brain development and adult synaptic plasticity [24]. As such, comprehensive genome-wide analyses of DNA and histone methylation remain uninvestigated as iron is a critical cofactor for DNA and histone modifying proteins, such as the ten-eleven translocation (TET) enzymes and the Jumonji C-terminal domain (JmjC) family of histone demethylases [25,26].

DNA methylation is essential for neuronal differentiation and maturation in the developing central nervous system and plays a critical role in learning and memory in the adult brain [27]. Altered DNA methylation patterns are associated with many neurological and psychiatric disorders [27]. While DNA methylation at promoter regions is relatively well studied and strongly associated with transcriptional silencing [28], methylation in intergenic regions and gene bodies has been less characterized and may have different functions [27]. Whole genome bisulfite sequencing (WGBS) is the most comprehensive method to analyze 5-methyl cytosine (5mC) at a single-nucleotide resolution [29]. In our previously published methodological paper, a novel method to identify differentially methylated regions (DMRs), namely the Defiant program, was developed. Here, using the same WGBS dataset [30], we present the first genome-wide assessment of DNA methylation in the developing postnatal day 15 (P15) rat hippocampus, during a period of peak iron deficiency and robust axonal growth and dendritic branching [18,31]. In addition to confirming previously reported individual genes and loci that were altered epigenetically due to iron deficiency, we identify novel loci critical to neural function that are epigenetically modified by early-life iron deficiency.

## 2. Materials and Methods

### 2.1. Animals

The University of Minnesota Institutional Animal Care and Use Committee approved all experiments in this study. Gestational day 2 (G2) pregnant Sprague-Dawley rats were purchased from Charles Rivers (Wilmington, MA). Rats were kept in a 12 h:12 h light:dark cycle with ad lib food and water. Fetal-neonatal iron deficiency was induced by dietary manipulation as described previously [32]. In brief, pregnant dams were given a purified ID diet (4 mg Fe/kg, TD 80396, Harlan Teklad, Madison, WI) from G2 to P7, at which time nursing dams were given a purified iron sufficient (IS) diet (200 mg Fe/kg, TD 01583, Harlan Teklad) to generate ID pups. Both ID and IS diets were similar in all contents with the exception of iron (ferric citrate) content. Control IS pups were generated from pregnant dams maintained on a purified IS diet. All litters were culled to eight pups with six males and two females at birth. Only male offspring were used in experiments.

### 2.2. Hippocampal Dissection

P15 male rats were sacrificed by an intraperitoneal injection of Pentobarbital (100 mg/kg). The brains were removed and bisected along the midline on an ice-cold metal block. Each hippocampus was dissected and immediately flash-frozen in liquid N_2_ and stored at −80 °C.

### 2.3. Whole Genome Bisulfite Sequencing and Library Preparation

Genomic DNA from IS and ID hippocampi was isolated using an AllPrep DNA/RNA Mini Kit (Qiagen). WGBS was performed using a previously published protocol [33]. Briefly, 1 µg of genomic DNA was fragmented into ~300 bp fragments using a M220 Covaris Ultrasonicator (Covaris, Woburn, MA, USA). Sequencing libraries were generated using a NEBNext genomic sequencing kit (New England Biolabs, Ipswich, MA, USA) and ligated with Illumina methylated paired end adaptors. Libraries were bisulfite-converted using an Imprint DNA modification kit (MilliporeSigma, St. Louis, MO, USA), and the size of 300–600 bp was selected using the Pippin Prep DNA size selection system (Sage Science, Beverly, MA, USA). Libraries were then amplified using Pfu-Turbo Cx Hotstart DNA polymerase (Agilent Technologies, Santa Clara, CA, USA). Paired-end libraries were sequenced to 100 bp on an Illumina hiSeq2000. Three biological replicates for each group were performed in WGBS. WGBS data are available on the Gene Expression Omnibus under GSE98064.

### 2.4. Identification of DMRs Using the Defiant Program

DMRs were identified by our in-house developed Defiant (DMRs: Easy, Fast, Identification and ANnoTation) program based on five criteria, as described previously [30]. Briefly, adapters were trimmed from the reads using a custom C language program. Trimmed reads were aligned against the rat genome (rn4). When reads overlapped at a base, the methylation status from read 1 was used. Methylation data at the C and G in a CpG pair were merged to produce the estimate for that locus. DMRs were defined with a minimum coverage of 10 in all six samples, *p*-value < 0.05, and a minimum methylation percentage change of 10%. Since the Defiant program did not use a pre-defined border to identify DMRs, the *p*-value < 0.05 cutoff only influenced the widths and quantity of DMRs. The Benjamini–Hochberg approach was applied for multiple testing to obtain false discovery rate (FDR, q-values). Genes were assigned to the DMRs based on a promoter cutoff of 15 kb to the transcription start site, with the direction of transcription taken into account.

### 2.5. Bioinformatics

The knowledge-based Ingenuity Pathway Analysis^®^ (IPA, Qiagen, Germantown, MD, USA) was employed to identify networks, canonical pathways, molecular and cellular functions, and behavioral and neurological dysfunctions using a P15 DNA methylation dataset from WGBS. The microarray dataset from a prior study [34] was also analyzed by IPA. IPA maps gene networks using an algorithm based on molecular function, cellular function, and functional group. Fisher’s exact test was used to calculate the significance of the association between genes in the datasets and the analyzed pathways or functions.

## 3. Results

### 3.1. Early-Life Iron Deficiency Induced Differential DNA Methylation in the Rat Hippocampus

We performed whole genome cytosine methylation bisulfite sequencing on P15 ID (n = 3) and IS (n = 3) rat hippocampi. To determine whether iron deficiency alters the genome-wide pattern of DNA methylation in the developing hippocampus, DNA methylation at 1000 randomly selected loci were compared between ID and IS samples to generate a representative heat map. This unsupervised clustering approach showed consistent patterns of methylation across all samples, without an overall shift toward hypo- or hypermethylation in the ID group (Figure 1a). To determine whether iron deficiency induces changes in DNA methylation at a locus-specific level, a ≥ 10% methylation change with *p*-value < 0.05 was used as an inclusion criterion [30]. We identified 229 DMRs (Figure 1b and Appendix A), including 58% intergenic, 26% intronic, and 11% exonic regions (Figure 1c). Approximately 4% of DMRs were located in promoter regions. These DMRs mapped to within 15 kb of the transcription start site of 108 genes with 63 hypermethylated and 45 hypomethylated loci in ID compared to IS hippocampi (Table 1).

### 3.2. Early-Life Iron Deficiency Altered the Methylation Status of Genes Regulating Neuronal Development and Function

To identify potential molecular pathways disrupted in the ID hippocampus, IPA was used to map DMRs onto functional networks. The top 10 canonical pathways are shown for DMRs from ID hippocampi (Table 2). Notable pathways critical for neuronal differentiation and function include β-adrenergic signaling, axonal guidance signaling, reelin signaling, Rho family GTPase signaling, cAMP-mediated signaling, and synaptic long-term potentiation.

### 3.3. The Methylation Status of Genes Regulating Axonal Guidance Was Altered in the P15 ID Hippocampus

Neuronal connections are formed by the extension of axons to reach their synaptic targets. This process is controlled by ligands and their receptors at the axonal growth cone, which can sense attractive and repulsive guidance cues to help navigate an axon to its destination [35,36,37,38,39]. These guidance molecules include netrins, slits, semaphorins, and ephrins. Iron deficiency altered methylation at the genes regulating ephrin B signaling/ephrin receptor signaling (data not shown), including increased methylation at *Ephb1*, *Itsn1*, *Prkar1b*, and *Srgap2*, and decreased methylation at *Arhgef15*, *Mknk1*, and *Slit3* loci (Table 1). Decreased methylation at *Arhgef15* and *Arhgef3* and increased methylation at *Map3k11* and *Ezr* loci suggest altered Rho GTPase signaling (Table 2), which transduces guidance signals in the growth cone and regulates cytoskeletal dynamics, an important cellular process for the formation of long-term potentiation (LTP) [40], a cellular basis of learning and memory [41,42].

### 3.4. Differential DNA Methylation is a Potential Epigenetic Mechanism Contributing to Neural Gene Dysregulation in the P15 ID Hippocampus

To determine whether differential DNA methylation in the P15 ID hippocampus potentially contributes to neural gene dysregulation, we compared our WGBS methylomic dataset and the P15 ID hippocampal transcriptomic dataset [34]. IPA revealed that cAMP-mediated signaling, axonal guidance signaling, reelin signaling, synaptic long-term potentiation, Rho family GTPase signaling, and ephrin B signaling were among the 18 pathways that were altered in both datasets (Table 3). The top functional networks altered in the P15 ID hippocampal methylome (Table 4) were also observed in the P15 ID hippocampal transcriptome. These include cell-to-cell signaling, nervous system development and function, behavior, neurological disease, molecular transport, and lipid metabolism. The transcriptomic dataset corroborates the methylome data and further highlights the disruption of synaptic transmission (Figure 2a), neuritogenesis (Figure 2b), and movement disorders (Figure 2c,d).

Integrating the P15 ID WGBS methylome and microarray datasets led to the identification of three genes, including *Pde2a*, *Mobp*, and *Cds1* (Table 5). All three genes showed differential methylation in their intronic regions. *Pde2a* (+28.6%) was hypermethylated while *Mobp* (−48.9%) and *Cds1* (−27.8%) were hypomethylated in the P15 ID hippocampus. All three genes were upregulated in the P15 ID hippocampus. While DNA methylation at gene promoters is strongly associated with gene silencing [28], DNA methylation in intronic regions may mark enhancers or repressors and can be associated with changes in gene expression [43,44]. Phosphodiesterase 2A (Pde2a) is highly expressed in the brain and metabolizes cGMP and cAMP to regulate short-term synaptic plasticity, axonal excitability, and transmitter release in the hippocampal, cortical, and striatal networks [45,46]. Myelin-associated oligodendrocyte basic protein (Mobp) is the third most abundant protein in the central nervous system (CNS), and is exclusively expressed in oligodendrocytes, the myelinating glial cells of the CNS [47]. Mobp plays a role in compacting or stabilizing the myelin sheath and regulates the morphological differentiation of oligodendrocytes [48]. CDP-diacylglycerol synthase 1 (Cds1) is a key enzyme in regulating second messenger phosphatidylinositol 4,5-bisphosphate (PIP_2_) levels. It is localized in the endoplasmic reticulum and mitochondria [49] and is involved in the synthesis of phosphatidylglycerol and cardiolipin, an important component of the inner mitochondrial membrane [50]. Cds1 is a novel regulator of lipid droplet formation, lipid storage, and adipocyte development [51], and plays a critical role in mammalian energy storage, which is compromised in developing iron-deficient neurons [52].

## 4. Discussion

Fetal and early postnatal life iron deficiency causes long-lasting impairments in learning, memory, and socio-emotional behaviors [1,14,15,16,53], including an increased risk for autism, depression, and schizophrenia in humans [2,54,55]. These long-term neurobehavioral deficits occur despite early diagnosis and treatment, indicating the need for adequate iron during critical periods of brain development. In preclinical models, these effects have been ascribed in part to the persistence of abnormalities in monoamine signaling, myelination, neural metabolism, and the expression of neuroplasticity-associated proteins into adulthood [20,56,57,58]. The molecular mechanisms underlying these persistent changes have not been fully elucidated. The present study goes well beyond previous studies by systematically analyzing the alteration of DNA methylation induced by early-life iron deficiency using a whole genome bisulfite sequencing approach. Consistent with previous transcriptomic analysis, the changes in DNA methylation in the ID hippocampus mapped to functional networks that are important for neuronal plasticity.

DNA methylation is an important epigenetic mechanism regulating gene expression, often across the lifespan. Methylation at genomic regions has different influences on gene transcriptional activity depending on the location of DNA methylation. In the present study, differential methylation was highly enriched at intergenic regions (58%) in the ID hippocampus. This outcome is similar to our previous findings in pancreatic islets of an intrauterine growth restriction rat model [59], where approximately 65% of DMRs were located in intergenic regions, as well as to other models of early-life adverse environments [33,60,61,62]. These conserved intergenic regions may represent important enhancers or cis-regulatory sites in regulating gene expression [43,63,64]. Thus, these DMRs might account for a substantially fewer number of loci with DMRs compared to a number of differentially expressed genes in the microarray dataset and a small overlap between these two datasets. Our data also showed that approximately 37% of DMRs in the ID hippocampus were located in gene bodies (26% and 11% in introns and exons, respectively). DNA methylation in gene bodies is generally associated with higher gene expression in dividing cells [65], in contrast to the regulatory effect of DNA methylation in promoter regions. However, this association is not seen in non-dividing cells [27]. Although not many cells in the hippocampus are actively dividing at P15, these DNA modifications might have occurred during the period of active proliferation in the prenatal period when the pregnant dam and fetus were iron-deficient. Additional DMR analysis at a timepoint when the developing hippocampus undergoes active proliferation will provide further insight into this notion. DNA methylation in gene bodies may define the exon boundaries, regulate alternative promoters in gene bodies, and regulate mRNA splicing and alternative splicing [65,66,67,68]. Wan et al. (2013) showed that tissue-specific DMRs are preferentially located in exons and introns of protein-coding genes [69]. These biologically relevant DMRs are enriched in alternatively spliced genes and a subset of developmental genes. It is possible that the real effect of DNA methylation in the P15 ID hippocampus is within these domains. Our microarray analysis [34] was insufficient to probe such effects. Finally, iron deficiency-induced intragenic DMRs could modify potential gene enhancers [43,44,70,71]. The intragenic DMRs in the ID hippocampus may directly contribute to neural gene dysregulation by modifying the accessibility of alternative splice sites or promoters. These analyses constitute a potential direction for future study.

Our WGBS analysis of the ID hippocampus identified pertinent signaling pathways that could underlie the neurobehavioral abnormalities associated with early-life iron deficiency. The DNA methylome showed that DNA methylation at genes regulating cAMP-mediated signaling and protein kinase A signaling was significantly altered in the P15 ID rat hippocampus (Table 2). Both pathways play critical roles in regulating LTP, as well as the plasticity of axonal guidance responses [72,73]. In addition, the predicted changes to the β-adrenergic signaling and nitric oxide signaling pathways in the ID hippocampus would likely result in lower activities of cAMP, cGMP, protein kinase C (PKC), mitogen-activated protein kinase (MAPK), and N-methyl-D-aspartate (NMDA) receptors [74,75]. Likewise, altered Rho GTPase signaling could change the axonal responses to guidance cues and affect neuronal connections and LTP formation. The Rho family of GTPases plays a key role in the formation of LTP by regulating cellular processes, including axon outgrowth and growth cone dynamics [76,77]. This effect is consistent with and provides a molecular basis for our previous finding of abnormal dendritogenesis and synaptogenesis in these ID rats [17]. Our study also revealed the alteration of the reelin signaling pathway in the ID hippocampus. Reelin regulates neuronal migration and cell positioning in the developing neocortex and cerebellar cortex [78], and modulates synaptogenesis, synaptic plasticity, and LTP, which are necessary for learning and memory in the adult brain [79,80]. Reelin can bind to two lipoprotein receptors, apolipoprotein E receptor 2 (ApoER2) and very-low-density-lipoprotein receptor (VLDLR), and initiates signaling cascades, including NMDA receptor activity, that are critical for hippocampal-dependent learning and memory [79,81]. Altered reelin signaling has been implicated in the pathogenesis of schizophrenia, bipolar disorder, and autism [82,83], all which have been associated with early-life iron deficiency [2,54,55].

DNA methylation patterns are dynamically regulated by DNA methyltransferase (DNMT) and TET enzymes [84]. TET proteins are responsible for catalyzing the conversion of 5-methylcytosine (5mC) to 5-hydroxymethylcytosine (5hmC) and other oxidized methylcytosines, thereby initiating the active DNA demethylation process. TET enzymatic activity is iron-dependent [25,85]. Consequently, iron deficiency could decrease TET activity, leading to global DNA hypermethylation. However, our WGBS results did not show an overall shift toward hypermethylation, suggesting a minimal effect of potentially compromised TET activity in the P15 ID hippocampus. As such, 5hmC has been shown to be an important epigenetic modification for chromatin structure and transcriptional regulation [86,87,88]. TET proteins and 5hmC are highly enriched in the brain and play an important role in neuronal development and differentiation [87,89,90]. Changes to 5hmC levels are associated with neurodegenerative diseases, such as Alzheimer’s disease, Huntington’s disease, and Parkinson’s disease [91]. Early-life iron deficiency may decrease TET activity and dynamically alter the DNA methylation pattern between 5mC and 5hmC, leading to the dysregulation of neuronal gene expression in the ID hippocampus. Due to methodological limitations, our current WGBS study could not distinguish 5mC and 5hmC levels. Investigations of loci-specific and genome-wide 5hmC alterations in the ID hippocampus are ongoing and will be presented in a future publication.

DNA methylation has been demonstrated to be an important factor in central nervous system development, the modulation of normal brain function, and the pathogenesis of neurological and psychiatric disorders [27]. Despite the limitations of our study, such as minimal biological replicates due to the high cost of WGBS and possible sex-dependent differences in ID rat brains, our current study is the first hippocampal methylome study on the developing rat brain, and provides evidence for DNA methylation as a potential epigenetic mechanism contributing to hippocampal gene dysregulation in early-life ID animals. Given that 95% of DMRs were found in the intergenic and intragenic regions, future studies will need to uncover the mechanisms by which DMRs reprogram gene regulation and mRNA splicing alteration in the ID hippocampus.

## Figures and Tables

**Figure 1 nutrients-11-01191-f001:**
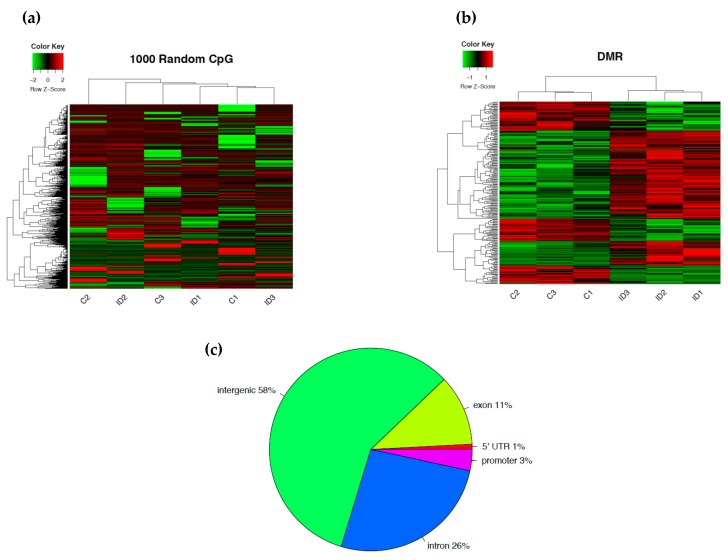
DNA methylome of the postnatal day (P) 15 rat hippocampus. (**a**) An unsupervised clustering heat map of 1000 randomly selected loci showing an absence of bias in global methylation between iron-sufficient (IS) and iron-deficient (ID) hippocampi. Each row in the heat map corresponds to data from a single locus. The branching dendrogram at the top corresponds to the relationships among samples. Hyper- and hypomethylation are shown on a continuum from red to green, respectively. (**b**) Heat map of differentially methylated regions (DMRs) showing significant differences in cytosine methylation between IS (labeled C1-3) and ID (labeled ID1-3) hippocampi. Each row in the heat map corresponds to data point from a single locus, whereas columns correspond to individual samples. The branching dendrogram corresponds to the relationships among samples, as determined by clustering using the 229 identified DMRs. Hyper- and hypomethylation are shown on a continuum from red to green, respectively. (**c**) Pie chart representing the location and proportion of DMRs. The gene body included exons and introns. The promoter was limited to 15 kb upstream from the transcriptional start site. The 5′-untranslated region began at the transcription start site and ended before the initiation sequence. The intergenic region is comprised of the regions not included in the above defined regions.

**Figure 2 nutrients-11-01191-f002:**
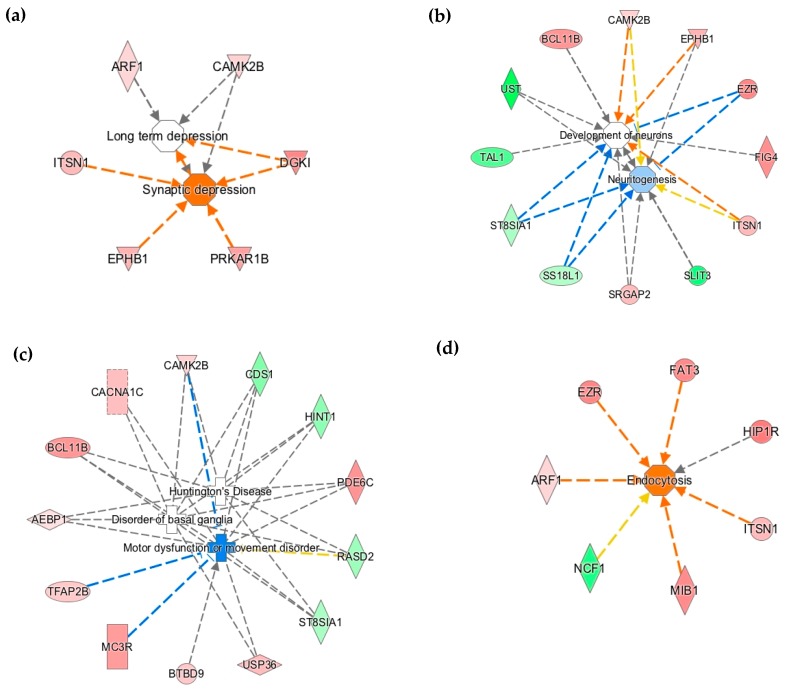
Ingenuity Pathway Analysis^®^ (IPA) functional annotation of altered DNA methylation at loci that are involved in (**a**) synaptic depression, (**b**) neuritogenesis and neuronal development, (**c**) pathogenesis of neurological diseases, and (**d**) lipid metabolism and molecular transport, including endocytosis. Red-filled and green-filled shapes indicate increased and decreased methylation, respectively. Orange-red lines indicate activation; blue lines indicate inhibition; yellow lines indicate findings inconsistent with the state of downstream activity; grey lines indicate that the effect was not predicted.

**Table 1 nutrients-11-01191-t001:** CpG methylation within the 15 kb promoter region of genes in the P15 iron-deficient rat hippocampus.

Hypermethylation				Hypomethylation			
Gene Name	#CpG	DMethylation(%)	*q*-value	Gene Name	#CpG	DMethylation(%)	*q*-value
*Adamts19*	5	58.5	0.016	*Abhd11*	5	−36.6	0.032
*Aebp1*	5	10.5	0.118	*Adarb2*	6	−19.7	0.031
*Ak4*	6	56.0	0.016	*Arhgap28*	5	−21.2	0.026
*Ankrd13a*	5	26.8	0.024	*Arhgef15*	5	−41.6	0.025
*Arf1*	6	17.8	0.026	*Arhgef3*	6	−19.8	0.048
*Arhgap31*	5	36.8	0.035	*Armc8*	9	−10.7	0.041
*Armc3*	8	43.1	0.031	*Bag2*	6	−21.8	0.059
*B4galnt3*	5	31.8	0.040	*Cds1*	6	−27.8	0.047
*Bcl11b*	6	44.7	0.035	*Commd1*	6	−40.4	0.016
*Btbd9*	5	23.6	0.039	*Dip2a*	10	−57.3	0.039
*Cacna1c*	5	27.9	0.032	*Dnaja2*	6	−39.0	0.037
*Camk2b*	9	19.3	0.032	*Dnpep*	5	−14.3	0.051
*Capn12*	7	21.5	0.024	*Dpf1*	10	−20.6	0.039
*Chd2*	7	59.7	0.031	*Dpf3*	6	−13.3	0.018
*Clvs1*	11	26.5	0.031	*Fkrp*	10	−77.7	0.018
*Cog3*	5	27.4	0.032	*Guca1a*	6	−29.0	0.032
*Dgki*	5	51.3	0.026	*Hint1*	8	−25.1	0.025
*Ephb1*	5	33.2	0.040	*Jak3*	8	−20.4	0.023
*Ezr*	5	51.9	0.018	*Kif26b*	6	−44.2	0.031
*Fat3*	9	50.1	0.018	*Klhl40*	5	−25.0	0.028
*Fig4*	5	47.9	0.016	*Lims2*	10	−11.6	0.016
*Foxb2*	5	32.6	0.039	*LOC691083*	5	−45.6	0.045
*Gucy2c*	9	86.6	0.016	*Mknk1*	5	−56.8	0.025
*Hip1r*	5	56.4	0.034	*Mobp*	5	−48.9	0.048
*Iqcg*	9	52.6	0.020	*Ncf1*	9	−38.4	0.032
*Itsn1*	5	29.8	0.032	*Pck1*	5	−48.2	0.023
*Jph3*	8	25.1	0.024	*Pgm3*	5	−69.0	0.024
*Kank3*	5	28.7	0.031	*Pon2*	6	−10.0	0.029
*Kctd15*	6	55.1	0.032	*Ppp1r3b*	7	−19.4	0.031
*Kctd6*	7	27.3	0.016	*Rasd2*	5	−22.4	0.016
*Macrod1*	5	35.4	0.033	*RGD735029*	5	−17.2	0.032
*Map3k11*	10	12.9	0.016	*Sardh*	5	−42.9	0.022
*Marveld2*	6	17.9	0.018	*Sh3pxd2a*	5	−39.4	0.016
*Mc3r*	5	42.2	0.016	*Slit3*	5	−41.5	0.029
*Mib1*	6	48.9	0.032	*Smyd3*	6	−28.2	0.112
*Mogat1*	5	27.4	0.035	*Ss18l1*	6	−18.6	0.042
*Mrpl19*	6	23.9	0.059	*St8sia1*	6	−20.1	0.016
*Myo3b*	5	17.1	0.024	*Tal1*	7	−35.1	0.024
*Neto2*	6	24.3	0.031	*Tmem120b*	5	−37.7	0.024
*Olr987*	5	21.1	0.016	*Tmem181*	5	−47.1	0.018
*Pabpn1l*	5	33.3	0.016	*Trrap*	5	−28.2	0.023
*Pde2a*	5	28.6	0.025	*Usf2*	10	−18.3	0.016
*Pde6c*	6	46.1	0.024	*Ush1g*	9	−17.2	0.037
*Ppp1r21*	15	24.9	0.030	*Ust*	5	−54.1	0.037
*Prkar1b*	5	40.2	0.024	*Wiz*	5	−45.3	0.034
*Ptpn14*	5	30.1	0.016				
*Rev3l*	5	16.7	0.023				
*Ric8b*	6	42.0	0.025				
*Riok2*	5	48.0	0.031				
*Sbk1*	6	39.5	0.026				
*Scrt2*	6	36.8	0.038				
*Slc38a1*	5	38.2	0.029				
*Slc5a1*	7	37.4	0.026				
*Snurf*	5	34.7	0.001				
*Spon1*	8	73.6	0.031				
*Srgap2*	5	28.6	0.026				
*Tbc1d20*	6	13.5	0.042				
*Tenm2*	5	42.5	0.016				
*Tfap2b*	5	22.6	0.035				
*Tgif2*	5	32.2	0.017				
*Tnni1*	31	29.8	0.016				
*Unc93b1*	5	54.8	0.020				
*Usp36*	21	28.8	0.043				

**Table 2 nutrients-11-01191-t002:** Top 10 canonical pathways implicated by DMRs in the P15 iron-deficient rat hippocampus.

Ingenuity Canonical Pathways	*p*-Value	Differentially Methylated Genes in the Pathway
Nitric Oxide Signaling in the Cardiovascular System	0.002	*CACNA1C*, *PRKAR1B*, *PDE2A*, *GUCY2C*
Cardiac β-Adrenergic Signaling	0.005	*CACNA1C*, *PRKAR1B*, *PDE2A*, *PDE6C*
cAMP-Mediated Signaling	0.005	*CAMK2B*, *MC3R*, *PRKAR1B*, *PDE2A*, *PDE6C*
Axonal Guidance Signaling	0.006	*ARHGEF15*, *ITSN1*, *SLIT3*, *MKNK1*, *PRKAR1B*, *EPHB1*, *SRGAP2*
Relaxin Signaling	0.007	*PRKAR1B*, *PDE2A*, *GUCY2C*, *PDE6C*
Reelin Signaling in Neurons	0.010	*ARHGEF15*, *ARHGEF3*, *MAP3K11*
G-Protein Coupled Receptor Signaling	0.011	*CAMK2B*, *MC3R*, *PRKAR1B*, *PDE2A*, *PDE6C*
Protein Kinase A Signaling	0.013	*CAMK2B*, *PTPN14*, *TNNI1*, *PRKAR1B*, *PDE2A*, *PDE6C*
Synaptic Long-Term Potentiation	0.021	*CAMK2B*, *CACNA1C*, *PRKAR1B*
Signaling by Rho Family GTPases	0.034	*ARHGEF15*, *ARHGEF3*, *MAP3K11*, *EZR*

**Table 3 nutrients-11-01191-t003:** Overlapping canonical pathways of the P15 DNA methylome and P15 microarray datasets.

	Methylome Analysis		Microarray Analysis	
Ingenuity Canonical Pathways	*p*-value	Differentially Methylated Genes	*p*-value	Differentially Expressed Genes
Nitric Oxide Signaling in the Cardiovascular System	0.002	*CACNA1C, PRKAR1B, PDE2A, GUCY2C*	0.000	*ITPR2, PIK3R3, KDR, PTPN11, PRKAA1, GUCY2D, ITPR1, CAMK4, PRKAG1, PDE2A, PDGFC*
Cellular Effects of Sildenafil (Viagra)	0.004	*CACNA1C, PRKAR1B, PDE2A, GUCY2C*	0.000	*MYH3, CACNG8, ITPR2, ADCY3, GPR37, GUCY2D, ITPR1, ADCY2, PLCE1, CAMK4, PRKAG1, PDE2A*
Cardiac β-Adrenergic Signaling	0.005	*CACNA1C, PRKAR1B, PDE2A, PDE6C*	0.036	*ADCY3, PKIG, ADCY2, PRKAG1, PDE2A, PPP2R2A, PPP1R11*
cAMP-Mediated Signaling	0.005	*CAMK2B, MC3R, PRKAR1B, PDE2A, PDE6C*	0.000	*GABBR1, CHRM3, CAMK4, VIPR1, PDE2A, Htr5b, CHRM2, CNGA2, CAMK2A, GNAI3, ADCY3, HRH3, PKIG, ADCY2, LHCGR, OPRM1, GRM6*
Axonal Guidance Signaling	0.006	*ARHGEF15, ITSN1, SLIT3, MKNK1, PRKAR1B, EPHB1, SRGAP2*	0.003	*CXCL12, PIK3R3, TUBB, EPHA3, ROBO1, PLCE1, DPYSL5, RTN4R, RTN4, GNAI3, FZD4, PDGFC, BAIAP2, SEMA4F, CXCR4, NRAS, CFL1, PTPN11, NTRK2, PRKAG1*
Relaxin Signaling	0.007	*PRKAR1B, PDE2A, GUCY2C, PDE6C*	0.008	*PIK3R3, ADCY3, PTPN11, GUCY2D, ADCY2, PRKAG1, PDE2A, NFKBIA, GNAI3*
Reelin Signaling in Neurons	0.010	*ARHGEF15, ARHGEF3, MAP3K11*	0.004	*PAFAH1B1, PIK3R3, PTPN11, APP, MAPT, ARHGEF9, APBB1*
G-Protein Coupled Receptor Signaling	0.011	*CAMK2B, MC3R, PRKAR1B, PDE2A, PDE6C*	0.000	*PIK3R3, GABBR1, CHRM3, CAMK4, VIPR1, PDE2A, Htr5b, NFKBIA, CHRM2, CAMK2A, GNAI3, NRAS, PDPK1, ADCY3, HRH3, PTPN11, ADCY2, PRKAG1, GRM5, LHCGR, OPRM1, GRM6*
Protein Kinase A Signaling	0.013	*CAMK2B, Ptpn14, TNNI1, PRKAR1B, PDE2A, PDE6C*	0.000	*ITPR2, PLCE1, NFKBIA, CNGA2, GNAI3, PYGB, ADCY3, PTPN11, ITPR1, ADCY2, PTPRF, TGFBR1, PPP1R1B, YWHAB, PPP1R11, DUSP12, PTPRN, CDC25A, PTPN2, PTPRO, H3F3A/H3F3B, CAMK4, PDE2A, PTPN23, CAMK2A, BAD, DUSP5, PTPN12, PRKAG1*
Breast Cancer Regulation by Stathmin1	0.017	*CAMK2B, ARHGEF15, ARHGEF3, PRKAR1B*	0.000	*ITPR2, PIK3R3, TUBB, CAMK4, PPP2R2A, CAMK2A, GNAI3, STMN1, NRAS, ADCY3, PTPN11, ITPR1, ADCY2, ARHGEF9, PRKAG1, PPP1R11*
Synaptic Long-Term Potentiation	0.021	*CAMK2B, CACNA1C, PRKAR1B*	0.000	*NRAS, ITPR2, GRINA, ITPR1, PLCE1, CAMK4, PRKAG1, GRM5, GRIN1, CAMK2A, GRM6, PPP1R11*
Gustation Pathway	0.023	*PRKAR1B, PDE2A, PDE6C*	0.000	*CACNG8, ITPR2, ADCY3, CACNB4, CACNA2D1, P2RX5, ITPR1, ADCY2, PRKAG1, PDE2A, P2RY1, CACNA1H*
Sperm Motility	0.023	*MAP3K11, PRKAR1B, PDE2A*	0.002	*ITPR2, PAFAH1B1, ITPR1, PLCE1, CAMK4, PRKAG1, PDE2A, CNGA2, CACNA1H*
GNRH Signaling	0.032	*CAMK2B, MAP3K11, PRKAR1B*	0.000	*CACNG8, ITPR2, CACNB4, CAMK4, CAMK2A, GNAI3, CACNA1H, NRAS, ADCY3, CACNA2D1, GNRHR, ITPR1, ADCY2, PRKAG1*
Signaling by Rho Family GTPases	0.034	*ARHGEF15, ARHGEF3, MAP3K11, EZR*	0.010	*BAIAP2, CFL1, RHOT2, PIK3R3, PTPN11, RHOB, CDH1, ARHGEF9, PLD1, RHOV, GNAI3, STMN1*
Molecular Mechanisms of Cancer	0.042	*CAMK2B, ARHGEF15, ARHGEF3, JAK3, PRKAR1B*	0.000	*RASGRF1, RHOT2, PIK3R3, CDC25A, CASP9, NFKBIA, CAMK2A, BAD, GNAI3, FZD4, NCSTN, NRAS, RALBP1, ADCY3, PTPN11, RHOB, HIF1A, ADCY2, CASP3, TGFBR1, CDH1, ARHGEF9, PRKAG1, RHOV*
Melatonin Signaling	0.048	*CAMK2B, PRKAR1B*	0.021	*PLCE1, CAMK4, PRKAG1, CAMK2A, GNAI3*
Ephrin B Signaling	0.049	*ITSN1, EPHB1*	0.022	*CXCL12, CXCR4, CFL1, CAP1, GNAI3*

**Table 4 nutrients-11-01191-t004:** IPA annotated functional similarity between the DNA methylome and transcriptome of the P15 ID rat hippocampus.

Category	Diseases or Functions Annotation	*p*-value	Differentially Methylated Genes	*p*-value	Number of Genes
Cell-To-Cell Signaling	Synaptic Depression/Neurotransmission	1.65E-04	*CAMK2B, ARF1, ITSN1, DGKI, PRKAR1B, EPHB1*	1.94E-10	21
Nervous System Development and Function	Neuritogenesis/Extension of Neurites	8.40E-03	*CAMK2B, ST8SIA1, ITSN1, SS18L1, BCL11B, EZR, SLIT3, UST, EPHB1, SRGAP2*	4.92E-16	62
Behavior	Locomotion	3.09E-04	*RASD2, HINT1, MC3R, BTBD9, NCF1, CACNA1C, JPH3, FIG4, TAL1*	1.08E-13	40
	Learning	2.22E-02	*CAMK2B, NCF1, BTBD9, DGKI, CACNA1C, PRKAR1B, JPH3*	3.51E-21	57
Neurological Disease	Cell Death of Cerebral Cortex Cells	1.33E-02	*ST8SIA1, ITSN1, MAP3K11, NCF1, SH3PXD2A*	8.55E-14	32
	Movement Disorder	4.68E-02	*CAMK2B, AEBP1, CDS1, ST8SIA1, HINT1, BCL11B, TFAP2B, PDE6C, USP36, RASD2, MC3R, BTBD9, CACNA1C*	5.58E-32	117
Lipid Metabolism	Quantity of Sphingolipid/Steroid	2.73E-03	*ST8SIA1, HINT1, BCL11B, PON2*	4.24E-09	40
Molecular Transport	Quantity of Heavy Metal	1.13E-02	*ARF1, USF2, COMMD1*	4.32E-19	58
	Transport of Molecule	1.92E-02	*SLC5A1, SLC38A1*	5.41E-31	144

**Table 5 nutrients-11-01191-t005:** Overlapping genes of the P15 DNA methylome and microarray datasets.

Gene Name	Symbol	∆ Methylation (%)	CpGs Location	FC (ID/IS)	Location	Type(s)
Phosphodiesterase 2A	*Pde2a*	28.6	Intron 2	1.16	Plasma Membrane	enzyme
Myelin-associated oligodendrocyte basic protein	*Mobp*	−48.9	Intron 2	1.37	Cytoplasm	other
CDP-diacylglycerol synthase 1	*Cds1*	−27.8	Intron 11	1.23	Endoplasmic reticulum & mitochondria	enzyme

∆ Methylation values are means from DNA methylome, and FC (fold change) values are means from microarray.

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
