# Peer review of "Dysregulation of Neuronal Genes by Fetal-Neonatal Iron Deficiency Anemia Is Associated with Altered DNA Methylation in the Rat Hippocampus"

_nutrients, 2019, doi:10.3390/nu11051191_

Round 1

Reviewer 1 Report

The authors have adequately address original comments. This paper is ready for publication. 

Author Response

We thank the reviewer for the comments and suggestions. We are very happy he/she likes our manuscript.

Reviewer 2 Report

This study has extended previous work to test the hypothesis that iron deficiency affects methylation in the hippocampus of the rat. The authors show that there are indeed differences, and attribute changes to functional alterations that are seen in adulthood.

I have no major concerns with he paper, but I would argue that the effect relates to iron deficiency rather than iron deficiency anaemia, as the authors state in the title. Unless they can provide good evidence that it is the anaemia, i think they should be more conservative and consider simply that it is iron deficiency that is generating the effect. 

Author Response

We thank the reviewer for the comment.

Although the effect was likely driven by ID (and we like this idea very much), but we could not rule out anemia’s role as these rat pups showed lower level of hematocrit (please see attached graph of our unpublished data). By definition, ID without anemia does not alter hematocrit level.

This manuscript is a resubmission of an earlier submission. The following is a list of the peer review reports and author responses from that submission.

Round 1

Reviewer 1 Report

This is a well-written manuscript with very intriguing results. DNA methylation and perinatal ID have not been well studied, and this research is a great first step in that area. A few changes are requested to add a few pertinent details for the readers before this can be published.

-Either in the methods or results section, more information should be provided on the data output from WGBS – any quality control checks used and coverage obtained per samples, at the very least.

-Ln 121: Should this be more than 4% of the 229 DMRs if they correspond to 108 genes?

-Make sure gene names are consistently italicized in Tables and text.

-In Figure 1c, proportions of DMRs in certain genic locations is shown. It would be helpful to put this into context with the proportions of those regions (e.g., intergenic, intronic, etc.) that all of the WGBS data (or at least all data with coverage >10) captured. This doesn’t have to be in the figure, but could be mentioned in the text.

-Table 1: It would be nice to see the p-values in this table, too. Also, were any of these DMRs <15 KB away from the promoter of multiple genes, and if so, how was the gene assigned here?

-In Tables 2 and 3, are the genes in ‘molecules’ the genes from the IPA analysis that were differentially methylated (or expressed) in that pathway? I assume so, but a more informative label instead of ‘molecules’ would help readers know for sure.

-Ln 330: Can you expand this to mention how inhibition of TET would be expected to alter DNA methylation patterns and if this is consistent with observed results (which are capturing total methylation, or 5mC + 5hmC)?

-Limitations of the study should be acknowledged somewhere in the discussion section. These limitations include comparing only 3 biological replicates from each group and using only male mice. It is also a limitation that p-values (not adjusted for multiple comparisons) were used throughout. This should be justified in the discussion section.

Reviewer 2 Report

Lien et al performed a methylomic pathway analysis on rat hippocampi isolated from rats previously exposed to iron deficiency in “Dysregulation of neural genes by fetal-neonatal iron deficiency anemia is associated with altered DNA methylation in the rat hippocampus”. Moreover, the authors provide an associative analysis in which they claim to correlate differentially methylated loci with gene dysregulation from a previously published transcriptome analyses. There are some serious issues that need to be solved before proceeding.

1.       The methylomic data analyses presented in the manuscript have been published before in reference [33], without appropriate citation. The data presented in the submitted manuscript are written in the abstract as if they are novel, but they are not!
E.g. citation from ref [33] “The DNA methylation profiles showed clear differences in DNA methylation between the iron sufficient and the iron deficient groups for regions near genes such as Pde6c, Chd2, Mobp, and Pck1.”
E.g.  citation from ref [33]  “The 229 DMRs that Defiant identified mapped within 15 Kbps of 108 genes (Additional file 1: Table S3). Among the 108 genes, 45 showed hypomethylated and 63 showed hypermethylated regions (Additional file 1: Tables S4 and S5, respectively). We identified that considerable portion of them (37 out of 43 hypomethylated and 31 out of 62 hypermethylated DMRs) were associated with neuronal function or development (Additional file 1: Tables S4 and S5)”
The data in Table 1 are published as supplemental table in ref [33]. It feels like self-plagiarism to me without appropriate citation.

2.       It is not clear from the references where the previous transcriptome analyses were published. The only thing new here might be the correlations.